# Evaluating the Performance of High Spatial Resolution UAV-Photogrammetry and UAV-LiDAR for Salt Marshes: The Cádiz Bay Study Case

Andrea Celeste Curcio [1,*] , Gloria Peralta [2], María Aranda [1] and Luis Barbero [1]

1   Department of Earth Sciences, Faculty of Marine and Environmental Sciences, International Campus of Excellence in Marine Science (CEIMAR), University of Cadiz, 11510 Puerto Real, Spain; maria.aranda@uca.es (M.A.); luis.barbero@uca.es (L.B.)
2   Department of Biology, Faculty of Marine and Environmental Sciences, International Campus of Excellence in Marine Science (CEIMAR), University of Cadiz, 11510 Puerto Real, Spain; gloria.peralta@uca.es
*   Correspondence: andrea.curcio@uca.es

**Abstract:** Salt marshes are very valuable and threatened ecosystems, and are challenging to study due to their difficulty of access and the alterable nature of their soft soil. Remote sensing methods in unmanned aerial vehicles (UAVs) offer a great opportunity to improve our knowledge in this type of complex habitat. However, further analysis of UAV technology performance is still required to standardize the application of these methods in salt marshes. This work evaluates and tunes UAV-photogrammetry and UAV-LiDAR techniques for high-resolution applications in salt marsh habitats, and also analyzes the best sensor configuration to collect reliable data and generate the best results. The performance is evaluated through the accuracy assessment of the corresponding generated products. UAV-photogrammetry yields the highest spatial resolution (1.25 cm/pixel) orthomosaics and digital models, but at the cost of large files that require long processing times, making it applicable only for small areas. On the other hand, UAV-LiDAR has proven to be a promising tool for coastal research, providing high-resolution orthomosaics (2.7 cm/pixel) and high-accuracy digital elevation models from lighter datasets, with less time required to process them. One issue with UAV-LiDAR application in salt marshes is the limited effectiveness of the autoclassification of bare ground and vegetated surfaces, since the scattering of the LiDAR point clouds for both salt marsh surfaces is similar. Fortunately, when LiDAR and multispectral data are combined, the efficiency of this step improves significantly. The correlation between LiDAR measurements and field values improves from $R^2$ values of 0.79 to 0.94 when stable reference points (i.e., a few additional GCPs in rigid infrastructures) are also included as control points. According to our results, the most reliable LiDAR sensor configuration for salt marsh applications is the nadir non-repetitive combination. This configuration has the best balance between dataset size, spatial resolution, and processing time. Nevertheless, further research is still needed to develop accurate canopy height models. The present work demonstrates that UAV-LiDAR technology offers a suitable solution for coastal research applications where high spatial and temporal resolutions are required.

**Keywords:** salt marshes; light detection and ranging (LiDAR); photogrammetry; multispectral; high resolution; unmanned aerial vehicle (UAV); digital models

## 1. Introduction

Salt marshes are highly complex systems with high ecological values and abundant ecosystem services [1,2]. Salt marshes protect coastal areas from floods and storms [3,4], prevent coastal erosion [5], store significant amounts of organic carbon [6], recycle nutrients, and remove pollutants, thus improving habitat quality and maintaining a high level of productivity in habitats with rich biodiversity [7,8].

Despite their importance, up to 70% of worldwide salt marshes have been lost in the 20th century [8], mostly due to extensive anthropogenic land cover changes that have accelerated marsh degradation. Climate change is now making these threats much more severe [5,9,10], with sea-level rise (SLR) being probably [8] the greatest current threat to salt marshes. Rising local sea levels could put salt marshes at risk of drowning, depending on SLR scenarios [11]. These habitats may compensate for the situation with natural mechanisms that maintain their elevation above local sea level [5,12]. These mechanisms include biophysical interactions between plants and soil and local sediment dynamics [12]. Nevertheless, natural events, such as inundations, and human activities, such as changes in land use [3,12,13], may destabilize these mechanisms, compromising [3,12,13] the ability of salt marshes to adapt to future SLR scenarios [14].

Fortunately, conservation efforts, such as the Ramsar Convention's implementation [15], have slowed the erosion of salt marshes during the past few decades. However, further efforts are still needed to improve the likelihood of salt marsh survival, requiring an interdisciplinary approach to understanding the underlying mechanisms [16,17]. Field databases required for modeling such processes need to be extensive (i.e., sediment availability, accurate topography, distribution, and vegetation productivity) and of high quality [11,18,19]. Due to the great accuracy needed for modeling coastal processes [20], the difficulties of accessing these environments, and the disturbance of the sediment during sampling, in situ monitoring is still problematic in salt marshes [21]. Techniques for remote sensing (RS) have proven to be an excellent tool for gathering spatial environmental data [22,23]. Traditional platforms include satellite or aerial systems, which are frequently utilized for many studies at the regional level, such as the mapping of tidal marshes [24], the monitoring of vegetation cover [25,26], and coastal management [27]. The spectral, spatial, and temporal resolution of satellite images is constrained, making them generally unsuitable for modeling ecological processes [28,29]. Unmanned aerial vehicles (UAVs) are bridging the high spectral, spatial, and temporal resolution gap left by satellites, enabling the development of rapid and affordable approaches. High-resolution photogrammetric cameras are among the current UAV sensors, while most other methods have also been effectively applied in conventional RS platforms (i.e., thermography, multispectral, LiDAR, and hyperspectral).

Three RS methods have a great potential for high-quality monitoring of salt marshes. (1) Photogrammetry successfully creates orthorectified maps (i.e., orthomosaics) and topographic products using structure-from-motion (S*f*M) methods [30,31]. (2) Light detection and ranging (LiDAR) gathers highly reliable 3D point clouds for high-resolution topography modeling and creates digital elevation models (DEM) from digital surface models (DSM) by point cloud classification techniques [32,33]. (3) Multispectral techniques offer useful data for vegetation mapping [34].

S*f*M photogrammetric methods based on UAVs have proven to be particularly effective in mapping marsh surfaces and calculating canopy height [35–37]. Airborne-LiDAR has been shown to enhance habitat classification for wetland areas [38–40]. However, a significant obstacle to mapping and modeling salt marshes is the accuracy of elevation data [20]. On the one hand, slight elevational variations (in the order of centimeters) can have a significant impact on plant zonation, which affects biomass and species distribution [41]. On the other hand, the precision of ground elevation measurements (field and LiDAR) below thick vegetation cover is limited by the uneven ground surface and extremely dense covers typical of salt marshes. [42,43]. The accuracy of LiDAR-derived DEM has been improved by up to 75% using custom DEM-generation techniques [42], lowering the root mean square error (RMSE) with specie-specific correction factors [44], by adjusting LiDAR-derived elevation values with aboveground biomass density estimations [45], or by integrating multispectral data during the processing [46].

LiDAR can identify tiny spatial scale structures, which is important for monitoring and modeling elements and features in irregular and dense canopy environments such as salt marshes, offering great potential for studying heterogeneous surface environments.

Grassland [47,48], forest, and agricultural vegetation monitoring [49,50] have all shown the effectiveness of UAV-LiDAR. Currently, the quality of mapping ground elevation and vegetation characteristics of salt marshes based on UAV-LiDAR technology has only been evaluated once [51], and the same goes for assessing the accuracy of UAV-LiDAR and UAV-photogrammetry in determining elevation and vegetation features in salt marshes [52]. It is worth mentioning that results from S*f*M-based photogrammetry techniques and LiDAR-based techniques can be compared because they are conceptually independent. Photogrammetric processing is based on the reconstruction of models from images, which involves interpolating what is not visible on the surface. The LiDAR is an active sensor whose laser beams can penetrate the spaces between features and pick up small details, thereby combining the 3D information of the scene into the model. Pinton et al. [52] demonstrated that LiDAR technology generates more precise salt marsh DEM and DSM in comparison to the results from photogrammetry-based approaches, and improves habitat classification. Nevertheless, more salt marshes with a wider range of vegetation heights and densities must be tested to determine this effectiveness. An assessment of the effects of flight settings on the laser beam penetration of salt marsh vegetation is also needed.

The salt marshes of Cádiz Bay Natural Park (CBNP) are an excellent example of Atlantic tidal wetlands in the south of Europe. In addition to being a RAMSAR site, SAC, and SPA, Cádiz Bay was designated as a Natural Park in 1994. This system is within an important bird migration route and the southernmost tidal wetland in Europe. Additionally, due to its geographic configuration and location, it is particularly susceptible to the impacts of climate change [53]. This makes the salt marshes of Cádiz Bay an excellent natural laboratory for the study of climate change effects on tidal wetlands.

The main goal of this study is to understand the performance of UAV technologies in salt marshes. We will assess the benefits and drawbacks of using UAV-LiDAR and UAV-photogrammetry to create precise digital models (DEM and DSM). The related spatial accuracy will also be evaluated. Additionally, the effectiveness of supplementary multi-spectral data on habitat classification will be assessed. The capability of canopy penetration and the accuracy of canopy height model estimations is explored by using several LiDAR sensor setups. Our findings will establish the conditions for standardizing the application of UAV technology in the study of salt marshes and will provide the first data for modeling the future responses of the Bay of Cádiz to SLR scenarios.

## 2. Materials and Methods

### 2.1. Site Description

The study area is located in Cádiz Bay, on the southwestern Atlantic coast of Spain (Figure 1). This bay also represents the southernmost example of the European coastal wetlands, right on the intersection between the Mediterranean and Atlantic oceans and the European and African continents. The most representative habitat of Cádiz Bay is the tidal marsh, presenting large extensions of this environment [54].

Cádiz Bay is divided into two waterbodies. A narrow strait connects an inner shallow basin, with a mean depth of around 2 m, with a deeper external basin, with depths up to 20 m and characterized by sandy beaches. The inner basin is a sheltered area protected from oceanic waves [55]. The study area is situated in the northwest corner of the inner basin (NE zone, 36°30′59.2″N 6°10′14.7″W), in front of the salina of San José de Barbanera. The intertidal system includes natural salt marshes, salinas, mudflats, and a complex network of tidal channels (Figure 1).

Cádiz Bay has a mesotidal and semi-diurnal tidal regime, with a mean tidal range (MTR) of 2.3 m, up to 3.7 m during spring tides [56]. The vegetation communities describe a typical salt marsh zonation of mid-latitudes [57], which can be divided into three main salt marsh horizons, depending on vegetation types and elevation ranges: upper, medium, and low marsh [54]. Unfortunately, in most cases, the upper marsh is interrupted by the protective walls of the salinas, with the most representative horizons of natural salt marshes of Cádiz bay being the medium and low ones. The medium marsh is dominated

by *Sarcocornia* spp. (mainly *S. fruticose* and *S. perennis*) and other halophytic species in lower abundance (Figure 2), and the low marsh is mainly dominated by *Sporobolus maritimus*. The lowest zones of the intertidal flats are colonized by sequential belts of seagrasses *Zostera noltei* and *Cymodocea nodosa* and small patches of *Zostera marina* [58].

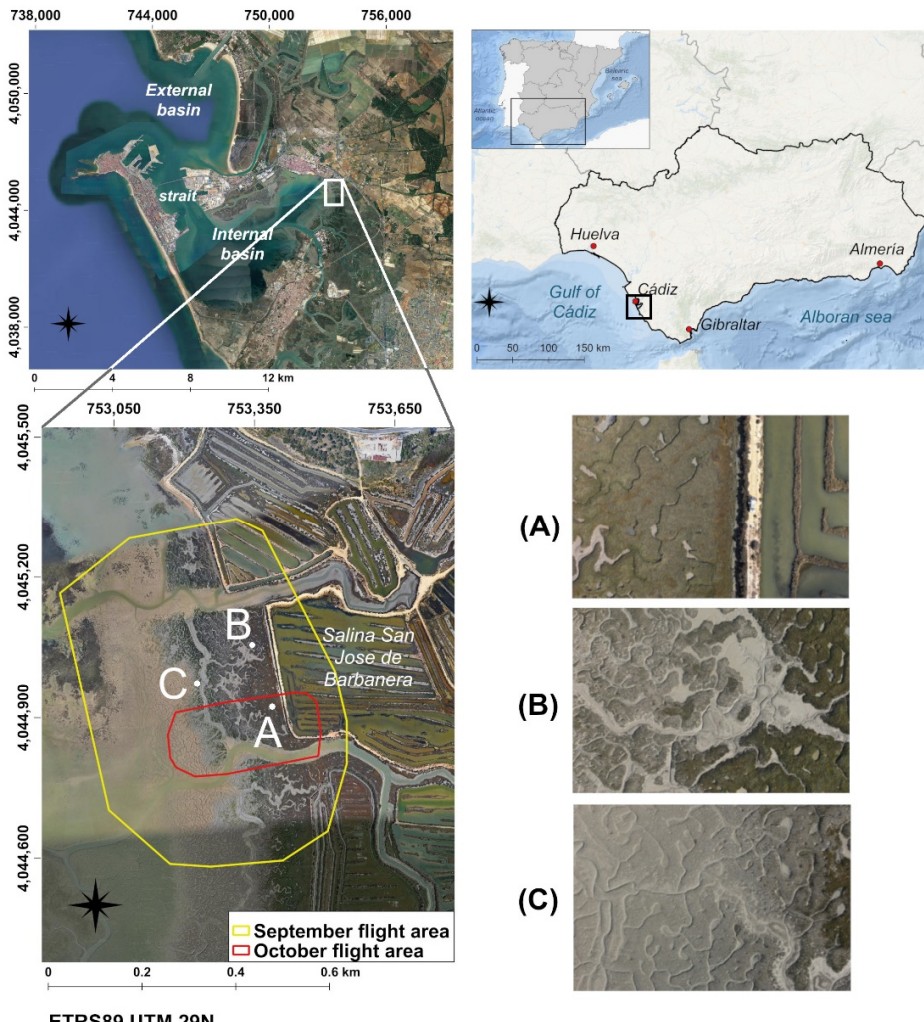

**Figure 1.** Location of Cádiz Bay and detailed view of the study site in the internal basin (in front of the salina San José de Barbanera). The yellow polygon indicates the flight area in September 2021 and the red one is for the flight area in October 2021. (**A**–**C**) are drone-captured images at the corresponding points in the left image: the uppermost part of the salt marsh system and a portion of a salina with its external wall on the right (**A**), a zone with a transition of dominant vegetation between *Sarcocornia* spp. and *S. maritimus* (**B**), and the lowermost part of the salt marsh system with a clear view of the tidal channel network (**C**).

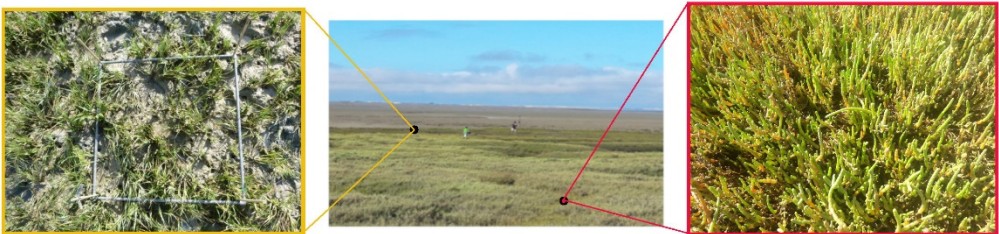

**Figure 2.** Westward view of the study area from the uppermost part of the salt marsh. The zoomed pictures show a detail of *Sporobolus maritimus*—predominant vegetation of the low marsh—(**left**) and *Sarcocornia* spp.—predominant vegetation of the medium marsh—(**right**).

The sampling site was selected according to the criteria of the width of the salt marsh vegetation and difficulty of access, as it was the best example of salt marsh plant zonation in the bay and was easy accessed by car. The conclusions of this work are expected to be of direct applicability to other tidal salt marshes across the world, given that low and medium tidal marshes usually present comparable structural properties [59].

### 2.2. UAV and Sensors

The drones service of the University of Cádiz (https://dron.uca.es/vehiculos-aereos/) (accessed on 25 April 2022) provided all of the equipment and sensors that were used for this work. The UAV used is a DJI Matrice 300 RTK quadcopter. The drone has an on-board RTK (real-time kinematic positioning) technology. The RTK records accurate GPS information during the flight, providing up to centimeter-level accuracy in geopositioning. The sensors implemented in the UAV were the photogrammetric sensor DJI Zenmuse P1, the DJI Zenmuse L1 LiDAR, and the Micasense RedEdge MxDual multispectral camera (Table S1). The missions were planned with the DJI pilot application.

The DJI Zenmuse P1 RGB photogrammetric supports 24 mm, 35 mm, and 50 mm fixed-focus lenses. For this work, the 35 mm fixed-focus lens was used, which, together with the 45 Mp full-frame sensor, provided an estimated value for ground sampling distance (GSD) of 1.26 cm/pixel. This sensor offers 0.03 m horizontal and 0.05 m vertical accuracy without deploying ground control points (GCPs).

The DJI Zenmuse L1 LiDAR sensor integrates a Livox LiDAR module, a high-precision IMU with a 20 Mp RGB camera with a focal length of 24 mm and a mechanical shutter on a stabilized 3-axis gimbal. Enabling the RGB camera entails collecting images, which can be used to assign the color to each point of the cloud generated by the LiDAR. When an adequate overlap is set, images can also be used to build an orthomosaic. The Livox LiDAR module has a maximum detection range of 450 m at 80% reflectivity. It can achieve a point cloud data rate of up to 240,000 points/s and it allows up to three returns per laser beam. The laser wavelength is 905 nm. The L1 LiDAR sensor supports two scan modes: repetitive and non-repetitive (Figure 3). The repetitive scan mode executes a regular line scan. The non-repetitive pattern is an accumulative process with an increase in the area scanned inside the field-of-view (FOV) together with the increase in integration time. This last pattern increases the probability of object detection within the FOV. The sensor can capture data from a nadir or oblique position. In a nadir flight, data are captured with the sensor axis oriented in a straight vertical position. The oblique flight configuration means data are captured with the sensor tilted at an angle with respect to the vertical. The sensor scans the area up to five times, changing the perspective from which the data are captured.

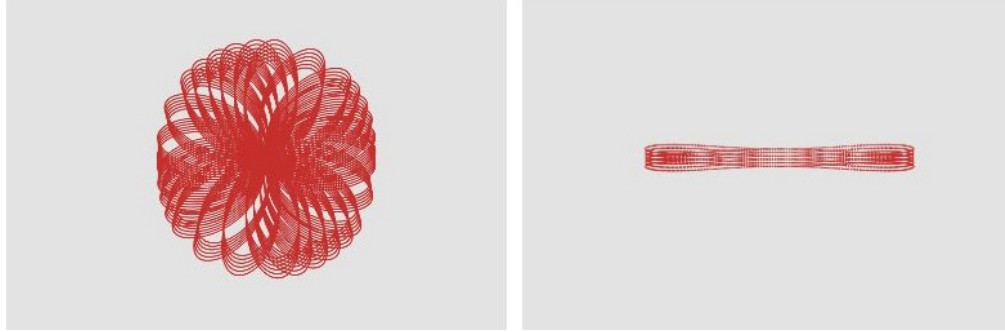

**Figure 3.** Non-repetitive petal scanning (**left**) and repetitive line scanning (**right**) of DJI Zenmuse L1 Livox LiDAR module.

The Micasense RedEdge-MX Dual sensor is a two-camera multispectral imaging system, with a total of 10 bands (five each camera), sampling data in the electromagnetic spectrum from blue to near-infrared. Two bands are centered in the blue (444 and 475 nm), two in the green (531 and 560 nm), two in the red (650 and 668 nm), three in the red edge (707, 715 and 750 nm), and one in the infrared (842 nm). The two-camera system is

connected to a downwelling light sensor (DSL2), which is used to correct for global lighting changes during the flight (e.g., changes in clouds covering the sun) and for sun orientation.

### 2.3. Flight Campaigns

The study area was surveyed in two consecutive campaigns at the end of summer and the beginning of autumn of 2021.

### 2.3.1. September Campaign

The first campaign was performed on the 8 September 2021, with a low tide of 1.4 m LAT. The campaign included three missions, covering an area of approximately 20 ha (yellow polygon, Figure 1). The first mission collected data with the photogrammetric sensor DJI Zenmuse P1, while the other two collected data with the DJI Zenmuse L1 LiDAR, changing some configurations in between missions (see Table 1 for mission configuration details). The two LiDAR missions were programmed with the repetitive scanning mode, double returns operating at a frequency of 240 kHz. The altitude of the LiDAR missions was set to obtain adequate point clouds, rather than an orthomosaic reconstruction. Nevertheless, the lateral overlap for the second LiDAR mission was increased to 70% to allow for the generation of the corresponding orthomosaic.

**Table 1.** Summary of flight configurations for the missions executed for this work.

| Date | Mission Name | Sensor | Covered Area (ha) | Flight Altitude (m AGL) | Side Overlap (%) | Frontal Overlap (%) | Speed (m/s) | Flight Time (min) |
|---|---|---|---|---|---|---|---|---|
| 8 September 2021 | P1 | P1 | 20 | 100 | 70 | 80 | 7 | 12 |
| 8 September 2021 | 100 m-L1 | L1 | 20 | 100 | 20 | n/a | 7 | 15 |
| 8 September 2021 | 60 m-L1 | L1 | 20 | 60 | 70 | n/a | 7 | 20 |
| 22 October 2021 | MS | MS | 4.5 | 100 | 70 | 80 | 3 | 5 |
| 22 October 2021 | 1–8 | L1 | 4.5 | 60 | 40 | n/a | 5 | * |

* Although using the same flight configuration, L1 flight time in the October campaign changed depending on the sensor configuration (see Table 2 for further details). Sensors: P1: photogrammetric sensor; L1: LiDAR; MS: multispectral.

**Table 2.** Configuration of LiDAR missions performed on 22 October 2021 in Cádiz Bay. The calibration column indicates whether the mission was performed before or after the vegetation was removed for the calibration trial. The flight time indicates the duration of the mission in minutes and seconds.

| Mission | Scan Mode | Sensor Orientation | Calibration | Flight Time |
|---|---|---|---|---|
| 1 | Non-repetitive | Nadir | Before | 2′48″ |
| 2 | Repetitive | Nadir | Before | 2′48″ |
| 3 | Non-repetitive | Oblique | Before | 18′30″ |
| 4 | Repetitive | Oblique | Before | 18′30″ |
| 5 | Non-repetitive | Nadir | After | 2′48″ |
| 6 | Repetitive | Nadir | After | 2′48″ |
| 7 | Non-repetitive | Oblique | After | 18′30″ |
| 8 | Repetitive | Oblique | After | 18′30″ |

### 2.3.2. October Campaign

The second campaign was performed on the 22 October 2021, with a low tide of 1.3 m LAT. This campaign included nine missions, one using the Micasense RedEdge MxDual sensor and eight missions with LiDAR (Table 1). The eight LiDAR missions only included four LiDAR configurations, but were duplicated to proceed with the calibration trial (Table 2; see Section 2.6).

The area covered in October was much smaller (4.5 ha approx., red polygon, Figure 1), but still representative of the system. The reduction was necessary to reduce the processing time for the collected multispectral data (MS).

The LiDAR missions had the aim of evaluating the best sensor setting combination for optimum accuracy/processing time balance. Settings evaluated included flight time, captured LiDAR data size, accuracy, and spatial resolution of deliverables. The sensor settings manipulated were scan mode (repetitive or non-repetitive) and sensor orientation (nadir or oblique) (Table 2). The missions were repeated for the calibration trial (see Section 2.6).

## 2.4. Data Processing

Orthomosaics are generated through photogrammetric processing of images, captured either by the Zenmuse P1 or the Zenmuse L1 LiDAR sensors. Digital models can be obtained from the photogrammetric processing of images or LiDAR processing of point cloud data (Figure 4). This section summarizes both types of processing, photogrammetric and LiDAR, as well as the methods to generate the multispectral masks and the digital models. Visualization and handling of raster deliverables were always done with the free and open-source software QGis.

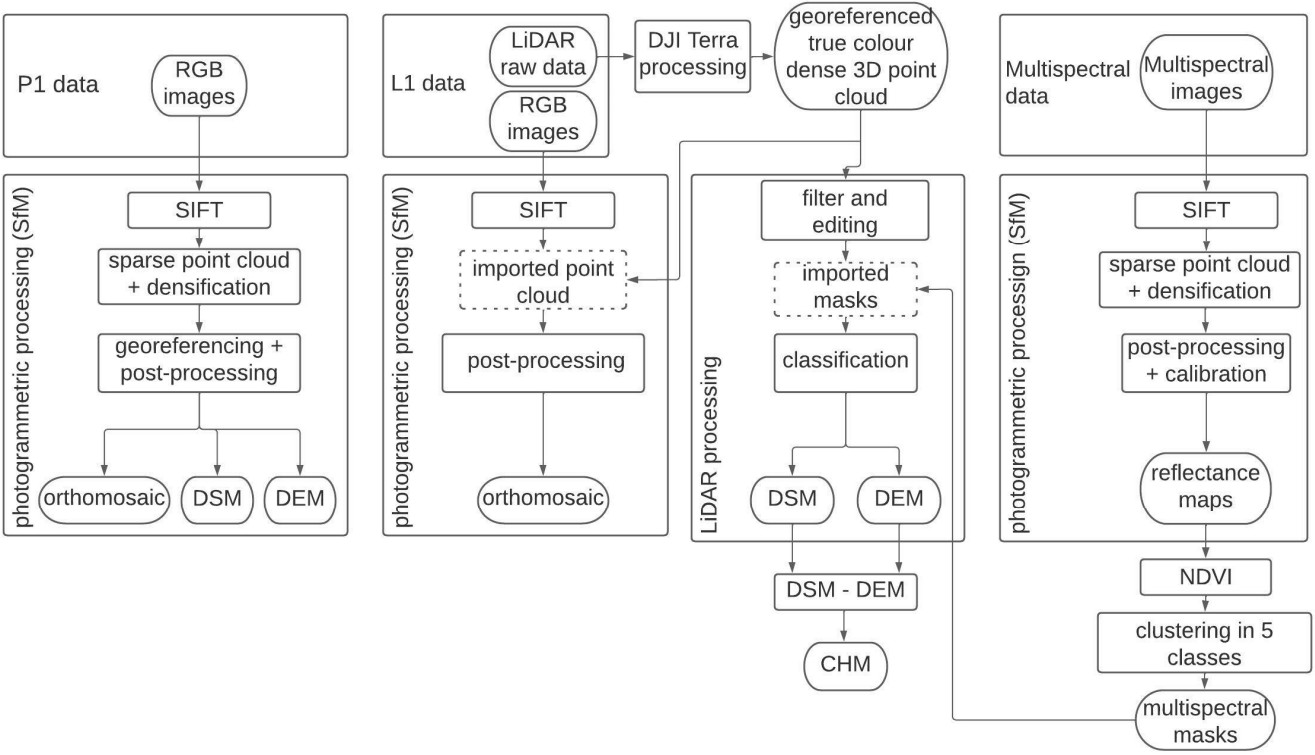

**Figure 4.** Flowchart showing the main data collected by the UAV sensors and their processing steps (photogrammetric camera on the left, LiDAR in the middle, and multispectral sensor on the right). The multispectral data allows the creation of masks for correctly classifying the plants in the point cloud. When multispectral imagery is missing, the imported mask step is absent in LiDAR data processing. Rectangular panels represent processing steps; rounded panels represent products.

### 2.4.1. Photogrammetric Processing

The Pix4Dmapper software [60], which transforms the images into orthomosaics and digital models, automatically implements the three steps of the structure-from-motion (S*f*M) algorithm workflow [30] (Figure 4, Table 3). In the first step, the scale invariant feature transform (SIFT) identifies key points from multiple images. The second step reconstructs a low-density 3D point cloud, based on camera positions and orientations, and densifies the cloud with the multi-view-stereo (MVS) algorithms. The third step is the transformation, georeferencing, and post-processing of the dense point clouds, producing the orthomosaics and the corresponding digital models. The ground sample distance (GSD) expresses the spatial resolution of the products in cm/pixel.

**Table 3.** Summary of processing operations and performing software as a function of the UAV dataset nature.

| Source | Dataset | Software | Process | End Product |
|---|---|---|---|---|
| P1 sensor, L1 sensor | Images | Pix4Dmapper | Photogrammetric processing (S*f*M) | Orthomosaic, DSM and DEM |
| L1 sensor | Images + 3D georeferenced and colored point cloud | Pix4Dmapper | Photogrammetric processing (S*f*M) | Orthomosaic |
| MS sensor | Multispectral images | Pix4Dmapper | Photogrammetric processing (S*f*M) and calibration | Reflectance maps |
| L1 sensor | L1 raw data | DJI Terra | First LiDAR processing | 3D georeferenced and colored point cloud |
| L1-DJI Terra-processed | 3D georeferenced and colored point cloud | CloudCompare | Point cloud comparison | Point cloud distances |
| | | Global Mapper LiDAR module | Filtering, classification, and digital model generation | DSM, DEM, CHM |
| CNIG | PNOA 2015-point cloud | LAStools | Digital model generation | DSM, DEM |

The Zenmuse L1 LiDAR sensor captures both image and point cloud datasets. Images can thus undergo photogrammetric processing to generate orthomosaic and digital models. However, for processing the RGB from the Zenmuse L1 LiDAR sensor, the second step of the S*f*M workflow is replaced by direct capture of the LiDAR 3D point cloud. Unfortunately, Pix4Dmapper does not allow for editing imported point clouds. Therefore, in those cases, the resulting DSM and DEM may contain larger errors and imperfections.

2.4.2. LiDAR Processing

DJI Terra software performs preliminary processing of the raw LIDAR data [61], which is required to produce a georeferenced, true color, dense 3D point cloud for the next steps (Figure 4, Table 3).

After pre-treatment, these datasets must also go through three major steps for processing (Figure 4), carried out using Global Mapper LiDAR module [62] (Table 3). Firstly, in order to increase the accuracy of the final products, the point cloud must be filtered and edited to remove artefacts and signal noises. With greater scan angles, a laser pulse travels a longer path, leading to biased measurements [63]. Therefore, the primary filtering method was to reduce the sensor's initial $-35°/35°$ range of scan angles to a proper range of $-26°/26°$. The classification of the points is the second phase. The algorithms employ geometric positions in relation to nearby points to assign the classes (see Digital Surface Models (DSM) Section). Vegetation masks can be created and imported into the procedure if multispectral data are available. By separating vegetated environments, these masks help with the accurate classification of plant points (see Section 2.4.3). The third step is the generation of digital models. By using data interpolation, this step reconstructs the ground surface, which results in the creation of the corresponding DEMs and DSMs.

The difference in elevation between DEM and DSM could be the height of the canopy, as there were no other items present apart from plants. Thus, using a geographic information system (e.g., Global Mapper or QGis), canopy height models (CHM) can be produced by subtracting one elevation model from another (see Canopy Height Models (CHM) Section).

Point Cloud Classification

An accurate DEM can only be obtained when the point cloud has been correctly classified. In our situation, classification entails designating each point to one of the following three categories: ground, non-ground, or noise. This method, in which the information on geometry and color is used to assign the class, is made possible by machine

learning algorithms. The method works effectively in contexts that are comparable to those used to train the algorithms (i.e., trees and buildings). The algorithm is not expected to operate efficiently in our study location, which is a flat, rough terrain with patches of low, dense vegetation. Manual intervention may be required, which can be a challenging and time-consuming operation.

The auto-classification tool recognizes noise and ground. The remaining points are labelled as non-ground points and interpreted as vegetation points.

Noise may be automatically identified with a classification algorithm that detects elevation values above or below a local average height. 'Maximum allowed variance from local average', and 'local area base bin size', which were set to 1 SD and 0.2 m respectively, are the input parameters for this algorithm. This means that using reference areas of 0.2 m, points with more than 1 SD of the local average height are classified as noise.

Ground auto-classification is done in two steps. The algorithm first determines non-ground points based on morphological attributes, such as the expected terrain slope and the ground's maximum elevation change. A second phase allows for the exclusion of some of those remaining from ground classification by comparing them to a simulated 3D curved surface representing the ground. The algorithm requires the neighboring area's size and the ground classification's vertical limit in order to compare the points [62].

The auto-classification process starts with default values that are then improved through trial and error. The parameters for the first filter were chosen based on the salt marsh's flat surface, with a maximum elevation change of 5 m and an expected terrain slope of 1 degree. The base bin size for modeling the 3D curved surface was set to 6-point spacing (ps). Two values of minimum height deviation from the local average height, 0.03 m and 0.10 m, were tested, in order to determine the appropriate threshold to differentiate vegetation from ground classification.

### 2.4.3. Masks from Multispectral Data

LiDAR data alone seems insufficient for high-quality classification of salt marsh point clouds. Hard and regular surfaces, such as roads, generate a single return LiDAR signal. However, salt marshes generate wide point clouds with scattered returns for the same LiDAR pulse. Thick point clouds in vegetated zones are reasonable and desirable for habitat classification. However, in salt marshes, bare grounds also produce thick point clouds, hindering the classification step (Figure 5). A method to solve this issue involves including additional information on the spatial distribution of vegetation. This information is incorporated into the process as multispectral masks that allow the vegetation zones to be separated from the bare ground ones, thereby allowing the creation of cut-off areas to successfully classify the point cloud.

The generation of multispectral masks requires the processing of the reflectance maps of the bands of the multispectral images. The procedure is similar to photogrammetric processing, except for the need for radiometric calibration. The calibration is done for each radiometric band, capturing the image of a calibration target immediately before and after the flight. The calibration target is made of a material with a known reflectance and allows the creation of reflectance-compensated outputs, in order to accurately compare changes in data captured over different days or at different times of day [64]. The Pix4Dmapper software calibrates and corrects the reflectance of the images according to the calibration values, delivering a total of 10 reflectance maps of the surveyed area.

The multispectral masks are obtained from the map of the normalized difference vegetation index (NDVI). The NDVI map is obtained by importing the reflectance band maps into QGis and stacking them together with the semi-automatic classification plugin (SCP) [65]; the NDVI was calculated according to Equation (1):

$$NDVI = (NIR - RED)/(NIR + RED) \qquad (1)$$

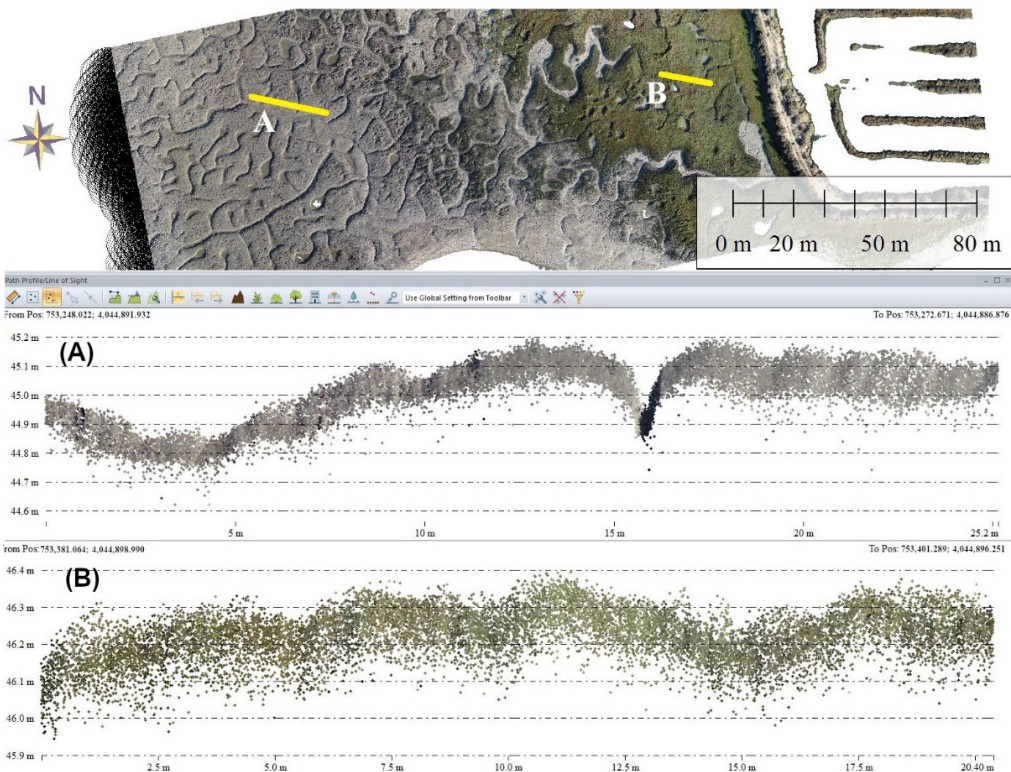

**Figure 5.** Profiles showing the thickness of the point cloud for bare ground (**A**) and vegetation (**B**).

Negative NDVI values correspond to water, while NDVI values close to zero represent the bare ground. Values higher than 1 correspond to vegetation, with values increasing with density and physiological conditions [25].

The NDVI raster can be classified using several clustering techniques. Among these, the 'k-mean clustering' technique was chosen for its quick and simple implementation. All it requires is to specify the number of clusters to generate; then, each object is placed in the cluster with the nearest "mean" [66]. The algorithms used were the combined minimum distance and the hill-climbing method, resulting in the definition of three classes (namely water, bare soil, and vegetation). The resulting raster is polygonized, and the classes are exported as separate shapefiles. These shapefiles are used for cutting the point clouds into vegetated and bare ground point clouds, treating each of them individually with different classification parameters. After that, the classified point clouds are merged into a single file. To validate the improvement provided by this method, classification results were corroborated visually. Furthermore, the proportions of vegetation and bare ground from each classified point cloud were compared to the values of coverage area obtained from the shapefiles. This would provide a rough estimate of the classification consistency.

### 2.4.4. Digital Model Generation

From P1 datasets, DSM and DEM are generated with the Pix4Dmapper software, whereas, for L1 datasets, the digital models are created using Global Mapper LiDAR software. The use of the LiDAR software in the second case is due to the limitations of the photogrammetric software. Pix4Dmapper lacks manual intervention options when point clouds are imported, leading to accuracy issues in the final products (see Section 2.4.1). All digital models are referred to as the ellipsoidal elevation.

#### Digital Surface Models (DSM)

When obtained from photogrammetric processing, DSMs were generated with the "Triangulation" method, which is based on Delaunay triangulation and recommended for flat areas [60]. When calculated from LiDAR data, the point clouds were manipulated

with the Global Mapper LiDAR module before the generation of the DSMs. In this case, the DSMs are generated with the binning method, a processing technique that takes point data and creates a grid of polygons, or bins [62].

Digital Elevation Models (DEM)

The DEM is the digital model resulting from excluding any feature on top of the ground after point cloud classification (Figure 4, Table 3, see Point Cloud Classification Section). For the specific case of P1 datasets, since the photogrammetric processing did not include point cloud classification, all points are treated as non-ground points, resulting in a DEM that is a smoothed version of the DSM.

DEMs are created only with points of the ground class. To identify the true ground points, the general practice is to use only the minimum values of the LiDAR point clouds. However, this method is inefficient in salt marshes, where true ground surfaces generate broad point distributions, underestimating the elevation of bare areas [42]. To address this specific problem of salt marshes, the true ground has been classified using the mean values of the cloud points instead of minimum ones.

Canopy Height Models (CHM)

Canopy height models (CHM) were generated by computing the DEM of difference (DoD), which is estimated as the difference between the DSM and the DEM. The result is a raster map with the canopy height distribution (i.e., CHM). This operation does not require matching resolution; it simply works based on cell overlap. Output resolution will be dictated by the element of the equation with the finest resolution (i.e., the DSM).

In order to determine whether UAV-LiDAR data can generate reliable CHMs, and test which is the optimal resolution of DSM and DTM needed to produce accurate estimates, DODs were generated by executing the subtraction operation using source digital models at different resolutions. Three DSMs—at 1, 3, and 5 ps resolution—and three DEMs—at 5, 10, and 15 ps resolution—were produced per LiDAR datasets. DoDs were generated using all possible combinations of DSM and DEM resolutions (i.e., the 1 ps DSM was subtracted from the 5 ps DEM, the 3 ps DSM was subtracted from the 5 ps DEM, etc.) for a total of nine DODs for each LiDAR mission.

*2.5. Accuracy*

RTK systems are supposed to be highly accurate. However, the P1 and L1 sensors have centimetric accuracy (see specifications in Section 2.2, Table S1). Therefore, it is necessary to quantify the accuracy of the products. For the accuracy of the products, the UAV sensor results are compared with ground control points (GCPs: blue, red, and yellow points in Figure 6A) measured in situ with a dGPS. For dGPS measurements, a LeicaGS18 GNSS RTK Rover was used, with horizontal and vertical measurement precision of 8 mm + 1 ppm and 15 mm + 1 ppm, respectively. In September 2021, the campaign included a total of 41 GCPs. Six of these GCPs were collected on the wall of the saline behind the sampling site. This provides a stable surface reference over time. In October 2021, the campaign included 63 GCPs (blue points in Figure 6A), with one of the GCPs on the wall of the saline and four GCPs at the calibration trial areas (two points per sector, yellow points in Figure 6B, Section 2.6). In this last campaign, the canopy height was also measured at the salt marsh GCPs.

Product accuracy was evaluated using the coefficient of determination ($R^2$) and root mean square error (RMSE), which can be calculated from the following Equations (2) and (3):

$$R^2 = 1 - \frac{\sum_{i=1}^{n}(x_i - y_i)^2}{\sum_{i=1}^{n}(x_i - x_m)^2} \tag{2}$$

$$RMSE = \sqrt{\frac{\sum_{i=1}^{n}(x_i - y_i)^2}{n}} \tag{3}$$

where n is the number of samples, $x_i$ and $y_i$ are the values from $i^{th}$ reference data (GCPs) and evaluated values (UAV sensor data), and $x_m$ is the mean of all reference data.

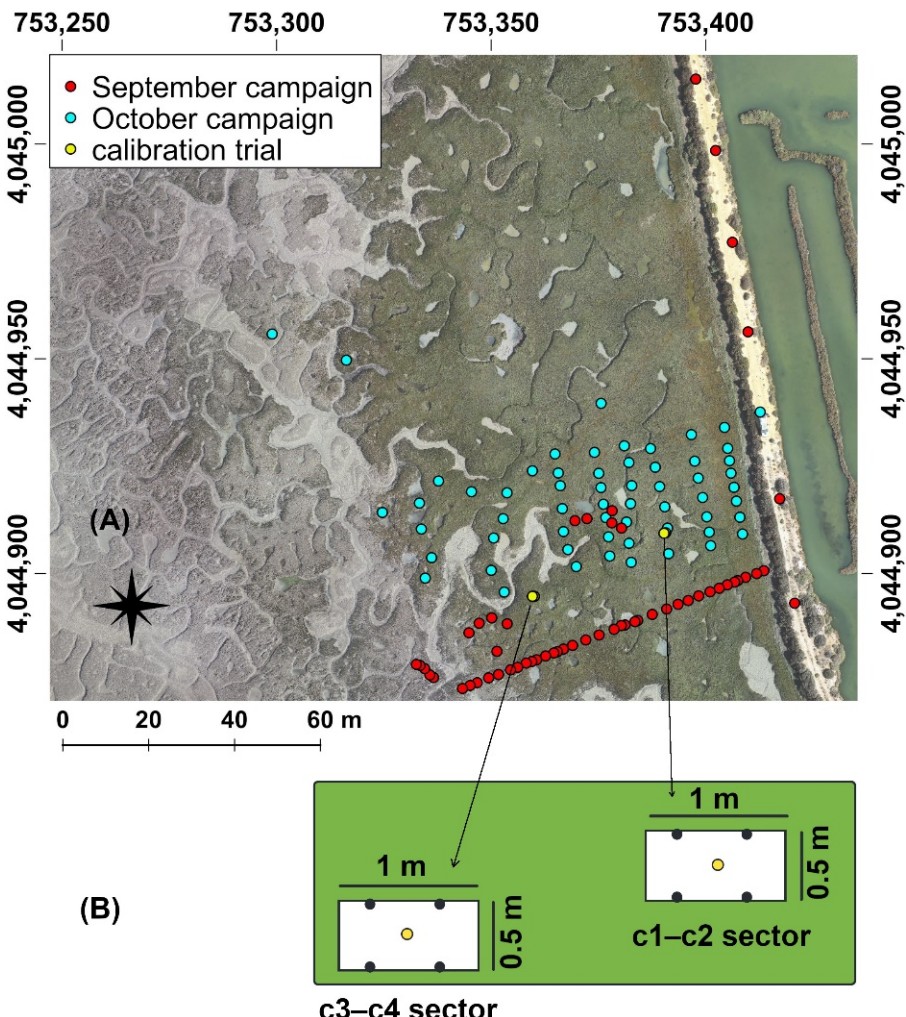

**Figure 6.** (**A**) Distribution of ground control points (GCP) over the surveyed area. The sectors for the calibration trial are identified by c1–c2 and c3–c4. (**B**) Trial operation scheme. White areas represent areas with vegetation removal. Black points indicate the location of canopy height measurements. Yellow points identify the location of dGPS measurements inside the trial areas.

The accuracy and mean errors of the orthomosaic were assessed through the photogrammetric software. This software estimates the position difference with respect to the GCPs. Only GCPs measured on the external wall of the saline were used for evaluating the photogrammetric processing reconstruction. To assess the quality of the point cloud, a linear regression between the dGPS measurements and their corresponding values in the point cloud was executed. The quality was evaluated with and without the saline wall GCPs.

In situ GCPs only contain information on ground elevation and canopy height (the last one only for the October campaign). Therefore, the accuracy of the digital models was only evaluated for DEM and CHM, but not for DSM (as we cannot obtain high precision field measurements of the landscape surface elevation).

### 2.6. Calibration Trial

To evaluate the potential of the UAV-LiDAR in discriminating ground and vegetation, a calibration trial was carried out in the October campaign. As part of the calibration study, aboveground vegetation was intentionally removed from two randomly selected

50 cm × 100 cm sections (yellow points in Figure 6A,B). The aboveground vegetation was pruned using garden shears. Then, the value of canopy height was measured at those plants present laterally at the four edges of the trial areas (black points in Figure 6B). Gathered values were used to estimate an average value for each sector, which was assumed to be representative of canopy height in those sectors. All October LiDAR missions were run twice, before and after the vegetation removal. Differences in elevation are expected to represent canopy height in the trial areas.

The LiDAR capacity for recognizing differences in elevation before and after vegetation removal was evaluated using three methods (see below). For each method, the goodness of fit was evaluated by comparing the field values with those obtained with the corresponding method.

### 2.6.1. Method A: Point Clouds

The first method compared LiDAR elevation data with field measurements. Vegetation and post-pruning datasets were compared using CloudCompare, an open-source 3D point cloud and mesh viewer and processing software [67] (Table 3). The distance between pre- and post-pruning point clouds was estimated with the 'Compute cloud/cloud distance' tool, using the 'Quadric' model and six neighbor points. This method allows for filtering and delimiting areas with height differences, sampling up to 20 points per area to estimate the corresponding value of the difference. The results were compared with the field measurements.

### 2.6.2. Method B: DSM

The second method compared the DSMs obtained from the missions before and after the calibration trial. This method evaluates vertical differences between pairs of DSMs. Up to 15 points per pair were sampled with the tool "Path profile" in Global Mapper and the value of the difference was estimated as the average of the 15 differences. The comparison was performed for all the DSM pairs, including photogrammetric and LiDAR-derived ones, evaluating the most reliable processing and resolution to detect canopy differences. The accuracy of the method was evaluated by comparing the results with field values, but also with points from the point cloud (Method A).

### 2.6.3. Method C: CHM

The validation of this method is done by applying the DoD to the calibration areas and cross-checking the results with field measurements of canopy height and point cloud-derived estimations. For this method, only flights before pruning were considered, comparing only the calibration areas.

### 2.7. PNOA 2015 Dataset

To evaluate the resources generated by the UAV-LiDAR, our data were compared with those of the LiDAR data of the Spanish National Plan of Aerial Orthophotography (PNOA). The PNOA provides a free library of orthophotography and LiDAR, the LiDAR resources having been initiated in 2009 (Centro Nacional de Información Geográfica—CNIG). This work required four PNOA 2015 LiDAR files since the area studied falls at the junction of four tiles of available point clouds (AND-SW, 214/216-4046/4048). These datasets were merged and cut to the same extent as our UAV missions and processed with the LAStools software [68]. Since the PNOA 2015 LiDAR dataset already comes classified, the corresponding DSM and DEM were generated without the classification step. The resolution and accuracy of the resulting digital models were compared with those of the UAV-LiDAR-derived results.

## 3. Results

### 3.1. Photogrammetric Processing Deliverables

From the Zenmuse P1 sensor, the photogrammetric deliverables include orthomosaics, DSMs (both with 1.25 cm/pixel GSD), and DEMs (6.25 cm/pixel GSD) (Table 4). From the LiDAR sensor, the orthomosaic resolution depends on the flight altitude. Surveys at 100 m had an average of 2.78 cm/pixel GSD. For LiDAR surveys at 60 m, the orthomosaic had a GSD of 1.69 cm/pixel. In general, photogrammetric processing is a very time-consuming task, with most of the time dedicated to densifying the point cloud. However, for L1 datasets, the processing is much shorter (Table 4), since this step is omitted due to the fact that the imported LiDAR point cloud is already densified.

**Table 4.** Characteristics of deliverables from photogrammetric processing (S*f*M) of P1 sensor and LiDAR sensor image datasets. Sens: sensor, Alt: flight altitude, Time: processing time for the entire project.

| Sens | Alt (m) | Captured Images | Data Size | Deliverable | Final Size | Resolution (cm/pixel) | Time |
|---|---|---|---|---|---|---|---|
| P1 | 100 | 374 | 8.34 GB | Orthomosaic | 2.98 GB | 1.26 | 6 h 30 min |
|  |  |  |  | DSM | 1.97 GB | 1.26 |  |
|  |  |  |  | DEM | 116 MB | 6.25 |  |
| L1 | 100 | 91 | 762 MB | Orthomosaic | 945 MB | 2.78 | 2 h 10 min |
|  | 60 | 236 | 1.92 GB | Orthomosaic | 1.26 GB | 1.69 | 3 h 45 min |

When evaluating the overall accuracy of the photogrammetric processing, the processing of P1 datasets generates products with a low general RMSE (0.044 m). The horizontal accuracy was comparable in both x and y coordinates (0.012, 0.009 m RMSE), but the error in the vertical dimension was higher (0.111 m RMSE). The processing of the L1 datasets generates products with RMSE even lower than P1 processing (0.006–0.010 m). The horizontal RMSE were <0.010 m and the vertical ones were 0.011 and 0.014 m for 100 and 60 m flights, respectively (Table S2).

The DEM produced by the photogrammetric processing of the P1 dataset has lower accuracy, with 0.335 m RMSE ($R^2$ 0.6027). The greatest deviations correspond to the six points located at the saline wall (Table S3). When such points are excluded, P1-derived DEM more accurately matches field measurements ($R^2$ 0.946, RMSE 0.070 m. Table S3, Figure S1).

### 3.2. LiDAR Processing Deliverables

LiDAR processing generates the full range of digital models (DEM, DSM, and CHM). Pre-processing the LiDAR point clouds requires 3D georeferencing and coloring the point cloud, which takes less than 10 min. The next step, filtering the effects of the scan angle, reduces the file size by up to 43% without changing the range of elevations. (Tables 5 and S18).

**Table 5.** LiDAR point cloud characteristics.

| Mission | Raw LiDAR Data Size | DJI Terra-Processed Data Size | Count Decrease after Filtering (%) |
|---|---|---|---|
| 100 m-L1 | 2.5 GB | 4.53 GB | 38.1 |
| 60 m-L1 | 3.7 GB | 7.65 GB | 39.7 |
| 1 | 270 MB | 415 MB | 17.1 |
| 2 | 270 MB | 430 MB | 42.7 |
| 3 | 1.99 GB | 3.5 GB | 15.8 |
| 4 | 1.99 GB | 2.6 GB | 39.6 |
| 5 | 270 MB | 410 MB | 17.0 |
| 6 | 270 MB | 430 MB | 41.9 |
| 7 | 1.99 GB | 3.2 GB | 14.6 |
| 8 | 1.99 GB | 2.4 GB | 36.6 |

Most of the salt marsh surface generated only one LiDAR return, precluding classification based on the number of returns. Elevation thresholds for ground/non-ground point classification proved to not be satisfactory. Depending on the threshold selected, ground points were either underestimated (0.03 m threshold) or overestimated (0.10 m threshold) (Tables S4, S5, S10, S11 and S20–S35). Multispectral data were used to resolve this problem. Using masks created from the multispectral dataset, point clouds could be classified much more accurately based on the actual distribution of the vegetation (Figures 7 and S7, Tables S36–S43). In general, any pattern described a dependence of classification performance on specific datasets. The estimations of the error vary arbitrarily, unaffected by flight type (nadir vs. oblique) or scan mode.

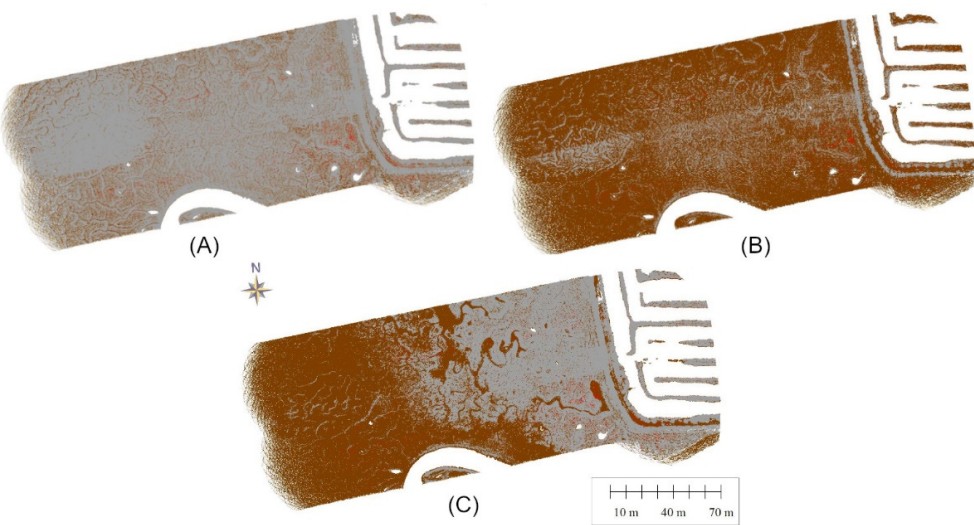

**Figure 7.** Approaches for LiDAR point cloud classification. (**A**) Classification after the application of the non-ground threshold of 0.03 m. (**B**) Classification after the application of the non-ground threshold of 0.10 m. (**C**) Classification after the introduction of multispectral data. Brown and grey points represent 'ground' and 'never classified' classes (i.e., vegetation), respectively. Red points are noise.

LiDAR point clouds had good accuracy, showing that the technology works well even when dealing with the challenges of salt marsh surfaces. Even when omitting stable GCP measurements (see $R^2$ value (marsh) in Table 6), the accuracy remains high, with $R^2$ values within 0.797–0.949 (Table 6). The lowest RMSEs corresponded to surfaces where LiDAR easily detects the ground (e.g., external wall). This was supported by the calibration trial, showing that the bare ground points in the pruned regions had the smallest deviations. On the other hand, the greatest RMSEs were associated with surfaces of *Sarcocornia* where vegetation obstructs LiDAR penetration (Tables S46–S53).

**Table 6.** Analysis of the LiDAR point cloud accuracy. The estimation of the RMSE includes all the GCPs of the corresponding campaign. RMSE: root mean square error. Marsh: including only GCPs on the marsh surface.

| Point Cloud Mission | RMSE (m) | $R^2$ Value | $R^2$ Value (Marsh) |
|---|---|---|---|
| 100 m-L1 | 0.115 | 0.976 | 0.939 |
| 60 m-L1 | 0.089 | 0.989 | 0.949 |
| 1 | 0.114 | 0.952 | 0.821 |
| 2 | 0.128 | 0.967 | 0.885 |
| 3 | 0.101 | 0.947 | 0.877 |
| 4 | 0.114 | 0.959 | 0.869 |
| 5 | 0.102 | 0.948 | 0.813 |
| 6 | 0.126 | 0.948 | 0.813 |
| 7 | 0.098 | 0.962 | 0.876 |
| 8 | 0.097 | 0.962 | 0.797 |

LiDAR Processing Digital Models

By flying at a lower altitude or using an oblique flight configuration, denser point clouds and products with finer resolution products can be produced, but at the expense of heavier datasets. The output resolution of the digital models depends on the density of the point cloud (Table 5) and the selected grid point spacing. For this task, the digital models were generated with up to three different point spacing values, obtaining DSMs with a resolution of 0.02–0.25 m/pixel and DEMs with 0.09–0.76 m/pixel resolution (Table S45).

LiDAR-derived DEMs had a high correlation with field data when using all available points ($R^2 > 0.86$). Correlation was lowered when using only points from the marsh surface (Table 7). As regards the sensor setting, the non-repetitive scan mode produced the most accurate DEM with lower RMSE. For the repetitive scan mode setting, DEM precision improves with a coarser resolution.

**Table 7.** LiDAR-derived DEM accuracy. nr: non-repetitive scan mode; r: repetitive scan mode; ps: point spacing.

| DEM Mission | Non-Ground Threshold | Average RMSE | | | $R^2$ Value | | | $R^2$ Value (Marsh) | | |
|---|---|---|---|---|---|---|---|---|---|---|
| | | 5-ps | 10-ps | 15-ps | 5-ps | 10-ps | 15-ps | 5-ps | 10-ps | 15-ps |
| 100 m-L1 | 0.03 | 0.088 | 0.086 | - | 0.970 | 0.973 | - | 0.867 | 0.881 | - |
| | 0.10 | 0.115 | 0.120 | - | 0.984 | 0.982 | - | 0.941 | 0.923 | - |
| 60 m-L1 | 0.03 | 0.097 | 0.140 | - | 0.971 | 0.930 | - | 0.972 | 0.971 | - |
| | 0.10 | 0.099 | 0.101 | - | 0.986 | 0.983 | - | 0.968 | 0.975 | - |
| 1nr | - | 0.069 | 0.056 | 0.076 | 0.904 | 0.930 | 0.888 | 0.703 | 0.774 | 0.768 |
| 2r | - | 0.099 | 0.096 | 0.064 | 0.873 | 0.904 | 0.969 | 0.593 | 0.682 | 0.916 |
| 3nr | - | 0.059 | 0.058 | 0.162 | 0.917 | 0.922 | 0.880 | 0.736 | 0.751 | 0.593 |
| 4r | - | 0.066 | 0.064 | 0.053 | 0.954 | 0.967 | 0.946 | 0.864 | 0.915 | 0.866 |
| 5nr | - | 0.064 | 0.067 | 0.069 | 0.924 | 0.916 | 0.916 | 0.813 | 0.809 | 0.846 |
| 6r | - | 0.100 | 0.091 | 0.065 | 0.863 | 0.903 | 0.949 | 0.611 | 0.723 | 0.859 |
| 7nr | - | 0.048 | 0.045 | 0.073 | 0.954 | 0.961 | 0.946 | 0.791 | 0.853 | 0.863 |
| 8r | - | 0.048 | 0.045 | 0.037 | 0.969 | 0.971 | 0.965 | 0.873 | 0.878 | 0.870 |

The nine combinations of DSM and DEM resolutions used for estimating CHM revealed that only the DSM resolution influenced the CHM results (Tables S62–S80). Therefore, only three CHMs per mission—those produced from the operation between DSM at three resolutions and the 5 ps DEM resolution—are displayed (Table 8). In general, the accuracy of the estimated CHMs was very low, with high RMSE (0.09–0.183 m) and low $R^2$ values (0.002–0.172, Table 8). These results show a lack of correspondence between modeled and field values.

**Table 8.** Analysis of canopy height model accuracy. The CHM resolution is indicated in the subindex with 0.02, 0.06, and 0.09 m/pixel, respectively. RMSE: root mean squared error.

| | $R^2$ | | | RMSE (m) | | |
|---|---|---|---|---|---|---|
| Mission | $CHM_{0.02}$ | $CHM_{0.06}$ | $CHM_{0.09}$ | $CHM_{0.02}$ | $CHM_{0.06}$ | $CHM_{0.09}$ |
| 1 | 0.002 | 0.004 | 0.005 | 0.123 | 0.156 | 0.173 |
| 2 | 0.038 | 0.029 | 0.043 | 0.179 | 0.167 | 0.155 |
| 3 | 0.069 | 0.069 | 0.033 | 0.138 | 0.103 | 0.100 |
| 4 | 0.061 | 0.116 | 0.079 | 0.164 | 0.136 | 0.126 |
| 5 | 0.021 | 0.038 | 0.027 | 0.124 | 0.100 | 0.093 |
| 6 | 0.035 | 0.036 | 0.034 | 0.183 | 0.169 | 0.160 |
| 7 | 0.105 | 0.078 | 0.082 | 0.151 | 0.119 | 0.104 |
| 8 | 0.079 | 0.126 | 0.172 | 0.152 | 0.121 | 0.110 |

### 3.3. Calibration Trial

### 3.3.1. Method A: Point Clouds

Depending on the configuration of the missions (Table 2), point cloud values had slight differences, but were very close to the field measurements, with a deviation of +/− 0.01 m (Table 9). This result validated the use of cloud points as reference values in further assessments.

**Table 9.** Deviations between field measurements and trial calibration values according to methods A, B, and C. For Method B, p4d corresponds to photogrammetric-processing-derived DSM and the rest for LiDAR-processing DSM, indicating the corresponding spatial resolution (ps: point spacing). For Method C, the CHM subindex indicates the spatial resolution. Values are in m.

| | | Method A | Method B | | | | Method C | | |
|---|---|---|---|---|---|---|---|---|---|
| Sector | Mission | Point Cloud | p4d | 1-ps | 3-ps | 5-ps | $CHM_{0.02}$ | $CHM_{0.06}$ | $CHM_{0.09}$ |
| c1–c2 | 1–5 | 0.01 | 0.00 | 0.00 | 0.00 | −0.02 | 0.05 | 0.09 | 0.09 |
| | 2–6 | 0.01 | −0.01 | 0.01 | 0.00 | −0.02 | −0.06 | −0.04 | −0.03 |
| | 3–7 | 0.00 | 0.11 | −0.01 | −0.01 | −0.01 | −0.03 | 0.02 | 0.04 |
| | 4–8 | 0.00 | 0.00 | 0.00 | −0.02 | −0.02 | −0.01 | 0.01 | 0.02 |
| c3–c4 | 1–5 | −0.01 | 0.00 | 0.02 | −0.01 | −0.03 | −0.03 | 0.01 | 0.04 |
| | 2–6 | −0.01 | −0.02 | 0.00 | −0.01 | −0.04 | −0.06 | −0.02 | −0.02 |
| | 3–7 | −0.01 | 0.10 | −0.01 | −0.01 | −0.01 | −0.09 | −0.02 | −0.02 |
| | 4–8 | −0.02 | −0.01 | −0.02 | −0.02 | −0.02 | −0.07 | −0.05 | −0.02 |

### 3.3.2. Method B: DSM

The estimates from the comparison of paired DSM (before and after vegetation pruning) deviated from field measurements within a range of −0.04 and 0.11 m (Table 9). On average, the overall concordance between estimations and field values was good. However, data captured with the nadir-non-repetitive configuration (Missions 1 and 5) produced the most accurate estimates (Table 9). The coarsest resolution (5 ps) tended to have the highest underestimation of canopy height, while 3 ps models generated the lowest deviation values.

Exceptionally, the estimations from Mission 3 (oblique-non-repetitive configuration) systematically overestimated the canopy height by 0.10 m. However, this effect was attributed to the signal loss of the RTK during the flight, which made a post-processing kinematic (PPK) treatment necessary to transform the raw dataset into an operable LiDAR dense point cloud. This operation caused a small z-shift in the reconstructed point cloud. This z-offset affects the results of the photogrammetric processing, which does not edit imported point clouds—but not the LiDAR processing, which allows the shifted point cloud to be corrected with other datasets.

### 3.3.3. Method C: CHM

In general, CHM estimations had no good correspondence with field values at the removed-vegetation-areas (Table 9), although for some missions the differences were reasonable (e.g., +/− 0.01 m). Data captured by nadir repetitive flights (Mission 2) showed similar differences in the two trimmed areas, always underestimating canopy height. For the rest of the missions, differences between CHM and field measurements did not have a consistent pattern with similar proportions of under- and overestimations.

### 3.4. LiDAR Sensor Optimum Settings

LiDAR sensor settings were manipulated to evaluate the best setting combination for optimum accuracy/processing time balance (Table 2).

### 3.4.1. Sensor Orientation

Sensor orientation includes nadir vs. oblique sensor configuration. This setting has major consequences on covered area, classification processing time, point cloud size, density, and deliverables resolution. Oblique configuration generates larger point clouds than the nadir one, increasing dataset size with the spatial range of the tilted sensor (Table S16). A larger dataset size implies increasing time processing for classification, but also the possibility of generating products with finer resolution.

### 3.4.2. Scan Mode

Scan mode (repetitive vs. non-repetitive) did not influence flight time or captured dataset size. However, the repetitive scan mode increases the occurrence of extreme values that need to be cleaned and filtered before the datasets are acceptable for modeling. The proportion of points lost during filtering is 36–42% for repetitive scan mode vs. 14–17% for the non-repetitive one (Table 5).

### 3.5. Comparison of PNOA 2015 and UAV-Based Data

The PNOA 2015-point cloud had a lower density than point cloud data captured from UAVs. The PNOA 2015 dataset has 0.80 samples/m$^2$, which represents 400 to 2600 times lower point density than UAV-derived datasets.

PNOA 2015-point cloud is an already classified product and it is available in RGB and IR colors. However, when digital elevation models were derived from this dataset, the obtained output resolution was extremely poor for environmental applications at high resolution (Figure 8). The DSM has a resolution of 4.4 m/pixel, while the DEM has 18 m/pixel.

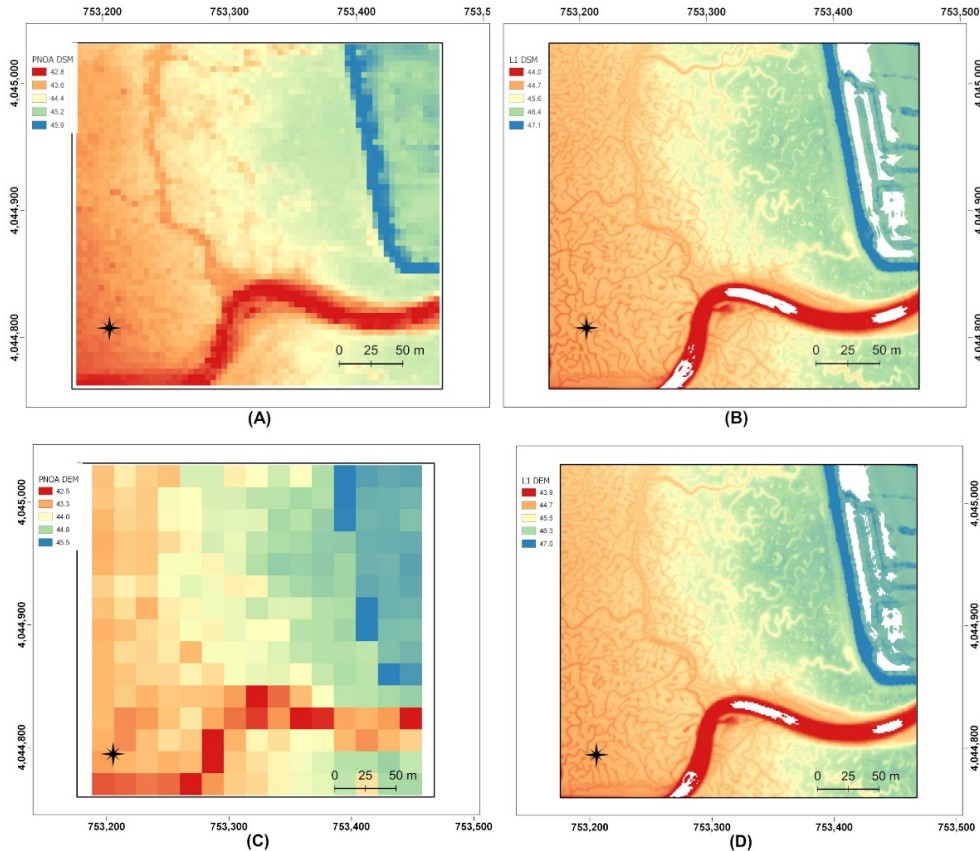

**Figure 8.** Comparison of PNOA 2015 and UAV-L1 products. (**A**) PNOA 2015 DSM. (**B**) 100 m-L1 DSM. (**C**) PNOA 2015 DEM. (**D**) 100 m-L1 DEM. Notice the great difference in resolution between PNOA 2015 and UAV-L1 products.

PNOA 2015 LiDAR data presented a large error in elevation values (1.36 m averaged RMSE), with individual point deviations between 1.01 m and 1.54 m. Nevertheless, the PNOA 2015 LiDAR data had a strong relationship with the field points ($R^2$ value 0.952 for all GCPs, 0.780 only for marsh GCPs).

PNOA 2015-derived DEM estimates elevations between 0.249 to 1.409 m below the field measurements (1.248 averaged RMSE). These results confirm that the PNOA 2015 LiDAR dataset underestimates local elevation of more than 1 m in the area of Cádiz Bay. The correlation between the PNOA 2015-DEM and field values was, nevertheless, strong ($R^2$ 0.885), albeit not as strong as the correlation between the UAV-LiDAR data and field values.

## 4. Discussion

### 4.1. Orthomosaic

Our UAV-photogrammetric system provides an extremely high detail orthomosaic, with a spatial resolution up to 1.25 cm. One drawback is that this high spatial resolution requires a long processing time since the image dataset from the photogrammetric sensor is four times larger than the corresponding LiDAR ones. Flying at the same altitude (100 m), the UAV-LiDAR system provides up to 2.7 cm spatial resolution, still very high for any salt marsh ground monitoring application. Orthomosaics of 2 cm/pixel resolution have been demonstrated to be particularly effective for the water and ecological environment monitoring, resulting in a better evaluation and development of hydrogeological simulation and temporal analysis of the area [69,70] The spatial resolution of our UAV-LiDAR products can be increased by reducing flight altitude (60 m), but to the detriment of the collected dataset magnitude, with heavier images and point cloud datasets.

Horizontal accuracy is crucial for detecting changes in spatial and temporal distributions with sea-level rise and when modeling the accretion or subsidence rate of salt marshes [10,11,71]. In our case, the processing of the photogrammetry images generates very high horizontal accuracy (1.2 cm, 0.9 cm for x and y coordinates). However, LiDAR images provide even higher horizontal accuracy (0.4 cm and 0.5 cm), demonstrating higher precision in positioning than photogrammetry images. The accuracies obtained here are higher than those found in studies in coastal areas, which reported accuracies ranging from 1.5 to 5 cm [35,72,73].

### 4.2. LiDAR Point Clouds

As expected, measurements at lower altitudes exhibited higher accuracy than at higher altitudes, but with small differences ($R^2$ 0.989 and 0.089 m RMSE vs. $R^2$ 0.976 and 0.115 m RMSE, respectively, Table 6). A lower altitude increases measurement reliability by increasing spatial resolution and point cloud density.

There is an average RMSE of nearly 0.10 m associated with all datasets, which supports the hypothesis of a systematic error. However, this is a reasonable value for LiDAR datasets, consistent with previous observations that ascribe up to 0.15 m of error to the limitation of the airborne laser to penetrate the dense vegetation [20,74]. Inaccuracy in salt marsh point clouds increases with the penetration issues in vegetation. The highest differences between LiDAR point cloud and field values were observed in areas covered by *Sarcocornia*, suggesting that this species generates the greatest issues of LiDAR penetration. This is not surprising, since this species forms very dense and thick shrubs covering the marsh surface like a carpet [75]. While technology can overcome this barrier, the best strategy is to determine the most appropriate model to explain the correlation between UAV-LiDAR and field data. Linear regression analysis seems to be a good approximation for salt marshes, especially if the dataset also includes measurements from stable surfaces (i.e., points collected on rigid structures stable over time).

### 4.3. Digital Models

The present study presents different types of digital models, including digital surface models (DSM), digital elevation models (DEM), and canopy height models (CHM). Both LiDAR and photogrammetric processing can provide digital models. Elevation values in photogrammetric sensor datasets are interpolated from point clouds generated from images, whereas the LiDAR sensor directly measures a point cloud of elevations. The main benefit of LiDAR is the capacity to penetrate spaces between features and pick up small details, whereas photogrammetry is limited to what is visible at the surface of the images.

### 4.3.1. Photogrammetric Processing Digital Models

The distribution of plants can be affected by differences of just a few centimeters, making maximum vertical accuracy crucial for salt marsh studies [76,77]. Digital models generated from our photogrammetric datasets did not offer the greatest vertical accuracy (RMSE 0.335 m). The high inaccuracy was related to the drone being unstable during the first flight line just over the points measured on the saline wall. When those points were excluded, field measurements and the P1-derived DEM appeared to match well (i.e., $R^2$ 0.946 and RMSE 0.070 m). This implies that the results should be interpreted cautiously if the drone's stability could not be ensured throughout the entire mission.

### 4.3.2. LiDAR Processing Digital Models

Point cloud classification is the most critical step in LiDAR processing for DEM generation. Initially, autoclassification algorithms were not able to correctly separate vegetation from ground points, although two elevation thresholds (0.03 m and 0.10 m) were tested for the classification algorithms. In terms of point cloud thickness, both vegetated and ground surfaces were comparable, making it difficult to classify them automatically. The rough and irregular surface of the dense canopy reduces the effectiveness of laser penetration, affecting the thickness of the point cloud corresponding to vegetated surfaces. Likewise, bare ground is not flat; it is irregular due to microtopography, rocks, vegetation remnants, puddles, etc., all of which produce scattering returns, which also increase the thickness of the point cloud. To overcome the limitation caused by the LiDAR point dispersion, a multispectral dataset was included. UAV-multispectral systems can be used to map plant communities in wetland environments with high accuracy [78,79]. In our case, NDVI-derived masks proved to be essential for the habitat classification allowing the adjustment of classification parameters and lower DEM error from an average RMSE of 0.11 m to 0.06 m (Tables S9, S15 and S61).

Modeling marsh environments requires high-quality elevation data. Alizad et al. [20] have shown that microtidal models are particularly vulnerable to imprecision because in these systems the error can be as large as the tidal amplitude. Instead, in mesotidal environments such as ours, with tidal amplitudes up to 3.6 m, an error of less than 20 cm (Tables S9, S15 and S61) can be considered irrelevant. A maximum error of 0.162 cm is observed in the produced DEMs, with an average error of 0.07 cm. The average error corresponds to 2.8% of the tidal amplitude (2.6 m), which proves the potential of the UAV-LiDAR system for the accurate elevation mapping of coastal marsh data.

Previous works have addressed the issues on DEM and DSM [80,81], associating the uncertainties and variability of digital models with surface complexity, field measurement accuracy, processing methods, interpolation, and resolution. These errors propagate when modeling DSM and DEM, resulting in amplified errors in the DoDs. These effects are supported by our results, explaining the low correlation between the CHMs and the field data.

Underestimations of canopy height on LiDAR-derived CHMs have already been documented [48,82]. Possibly, this issue is caused by insufficient laser scanning frequency for corresponding drone speeds, which translates into the missing vegetation tops [50,83,84]. The loss of information from the top of the canopy is more frequent than the loss of canopy bottom points, and it is mainly influenced by the maximum vegetation height, its standard

deviation, and true flight height [79]. This could explain why salt marsh DEMs are accurate, but CHMs are so inaccurate. Other interpretations could be that LiDAR technology does not provide accurate DSMs in salt marshes, or that field canopy height measurements are still inaccurate.

### 4.4. Calibration Trial

The calibration trial aimed to understand the sensor sensitivity in order to identify changes in elevations by comparing the mission data before and after the removal of vegetation. Method A contrasts paired point clouds (pre- and post-pruning) and shows a good correspondence between the difference in elevation computed from point clouds and field observations (deviations of 0–0.02 m). This is consistent with prior assessments of the efficiency of UAV-LiDAR in determining elevation changes [48,79,84].

Method B shows that paired DSMs and onsite values mostly agree, but that, as previously demonstrated [44], the interpolation of point clouds to generate digital models may introduce additional elevation error. Our findings support earlier findings that the higher inaccuracy for high-resolution models is related to the natural surface complexity of the environment. [80]. From the three resolutions tested (1, 3, and 5 ps), the intermediate (3 ps) seems to be the most appropriate value for obtaining the most reliable results.

Method C was unable to find a relation between values extracted from CHM and onsite measures at peeled areas. Method C was thus inadequate to validate sensor sensitivity. This experiment revealed that CHMs are inaccurate, not only in areas covered by dense vegetation (see Section 4.3.2)—which could limit laser penetration—but also in areas where the bare ground information is collected (i.e., peeled areas). This result supports the conclusion that interpolating CHMs from UAV-LiDAR may smooth the range of plant height and result in low accuracy of height-related structural features [79].

### 4.5. LiDAR Sensor Optimum Setting

The evaluation of sensor settings included the assessment of the sensor orientation (nadir vs. oblique) and scan mode (repetitive vs. non-repetitive). The sensor orientation produces very different flight and processing times. In our case, the oblique flights were six times longer and the datasets six to eight times larger than the nadir ones (Table 2). Our results agree with previous results that found that oblique flights improve accuracy when collecting point clouds, resulting in a high precision 3D reconstruction [85] (Table S54). However, the differences with accuracy from the nadir ones are very small. Therefore, the nadir configuration was considered preferable, as the level of detail of the products is adequate, but the datasets are much smaller and need less time for processing.

The scan mode includes repetitive and non-repetitive modes. The repetitive mode produces a higher occurrence of extreme scan angle points, which is the main factor in producing artefacts in the resulting models. This is consistent with the findings of Ma et al. [83], who revealed that as scan angle exceeded a specific threshold, the uncertainty in LiDAR-derived estimations increased significantly. Extreme scan angle points need to be removed, causing an important decrease in point counts and dataset density in datasets from repetitive scan mode flights. Non-repetitive scan mode generates more accurate DEMs (lower RMSE values) at a finer resolution. On the other hand, repetitive scan mode datasets require a coarser resolution to improve the precision of the DEMs. This is consistent with previous studies that explain this effect as a function of the data collection method: the repetitive scan mode works with a linear pattern, which is more sensitive to properties such as shininess, clarity, and color, resulting in larger variability and errors [86,87], whereas the non-repetitive mode improves the detection and details of objects, suggesting that this scan mode is the most suitable for salt marsh systems.

### 4.6. Comparison of PNOA 2015 and UAVs-Based Data

UAV-LiDAR technology is certainly a very effective tool in environments that require very high temporal and spatial resolution for accurate knowledge of the system, such as

salt marshes. PNOA datasets are freely accessible and cover the entire Spanish territory, but they have a very limited temporal and spatial resolution to use for modeling salt marsh processes. Our UAV-LiDAR sensor provided more detailed elevation information than any dataset available in the PNOA 2015 LiDAR library, with higher resolution, correlation, and accuracy. It is important to note that while UAV-LIDAR cannot capture data at the regional scale as a single mission, its high mobility and ease of use allow this technology to capture data for areas larger than the one of this study by planning several flights that, when stitched together, will provide a large spatial coverage. This will enable the use of this method in additional research contexts and environments where high temporal and spatial resolutions are necessary for monitoring programs.

The results presented in this study are in line with those presented by García-López et al. [88], who improved the previous PNOA-derived cartography through significantly reducing the spatial resolution of the mapped area by generating a DEM from a UAV-LiDAR dataset (0.069 m vs. 5 m). Our results also revealed a systematic underestimation of −1 m elevation in PNOA 2015. This corresponds to 44–48% of the mean tidal range of the area, which is definitely too high for high-resolution modeling of the system. Systematic error in other national-LiDAR datasets has been previously reported [42,89]. The authors attributed this error to the type of land cover surveyed and the physical and technological limitations of the employed LiDAR system. The PNOA 2015-derived digital models obtained underestimated the elevation values of the area in concordance with the findings of García-López et al. [88], who demonstrated a PNOA-LiDAR systematic error of −0.4 m for the marshes of Cádiz Bay.

The results from this work therefore reiterate that PNOA-LiDAR datasets can be useful for a first assessment and a general framework, but they should not be used for applications that require high precision, such as flood risk and coastal hazard estimates.

## 5. Conclusions

This study demonstrates the potential of UAV sensors for the study of complex and difficult access systems such as salt marshes, where inaccuracies are still difficult to overcome. Photogrammetric and LiDAR techniques provide orthomosaics and digital models at very-high spatial resolution. The LiDAR sensor can also capture images that generate products with high accuracy from lighter image datasets in a shorter processing time than the photogrammetric sensor (P1). Nevertheless, the photogrammetric sensor can provide a higher spatial resolution that can be an excellent complementary tool for limited areas. For the use of LiDAR in salt marshes, the nadir non-repetitive configuration seems the best setting for reliable results at fine resolution, providing the best balance between dataset size, spatial resolution, and processing time. Nevertheless, the best results require multispectral data to help with the discrimination of vegetated and non-vegetated zones. Our results demonstrate that LiDAR data can generate accurate salt marsh DEMs, suggesting that LiDAR can penetrate dense vegetation to some extent. However, unless the penetration and reflectance issues observed on natural salt marsh surfaces are solved, additional technical improvements are still required to generate reliable salt marsh canopy height models (CHMs). The inaccuracy of CHMs could be associated not only with LiDAR penetration issues, but also with the reliability of the ground truthing measurements in elevation and canopy height, as field measurements are challenging in this environment. UAV-LiDAR datasets can reach resolutions and accuracies unachievable from the datasets of the national cartography LiDAR library (PNOA-LiDAR 2015), which definitely have high applicability in large-scale frameworks, but lack the precision and details required for coastal research where high spatial and temporal resolutions are crucial. The results from this research can be used to plan monitoring programs in any marsh environment, as well as in other coastal and continental habitats.

**Supplementary Materials:** The following supporting information can be downloaded at: https://www.mdpi.com/article/10.3390/rs14153582/s1.

**Author Contributions:** Conceptualization, M.A., A.C.C., L.B. and G.P.; methodology, A.C.C. and L.B.; software, A.C.C.; formal analysis, A.C.C.; investigation, A.C.C.; resources, A.C.C. and L.B.; data curation, A.C.C.; writing—original draft preparation, A.C.C. and M.A.; writing—review and editing, A.C.C., G.P., M.A. and L.B.; visualization, A.C.C. and G.P.; supervision, G.P. and L.B.; project administration, A.C.C., L.B. and G.P. All authors have read and agreed to the published version of the manuscript.

**Funding:** This research received no external funding.

**Data Availability Statement:** The data presented in this study are openly available in Zenodo at https://doi.org/10.5281/zenodo.6850188. Publicly available datasets were analyzed in this study. This data can be found here: https://centrodedescargas.cnig.es/CentroDescargas/index.jsp (accessed on 24 January 2022).

**Acknowledgments:** The authors want to thank all the members of the drone service of the University of Cádiz, which provided all the UAV systems used to carry out the research for this study. The drones service of the University of Cádiz was equipped through the "State Program for Knowledge Generation and Scientific and Technological Strengthening of the R+D+I System State, Subprogram for Research Infrastructures and Scientific-Technical Equipment in the framework of the State Plan for Scientific and Technical Research and Innovation 2017–2020", co-financed by 80% FEDER project ref. EQC2018-004446-P. Reviewers and editors are acknowledged. The authors acknowledge the Program of Promotion and Impulse of the activity of Research and Transfer of the University of Cadiz for the productivity associated with the work. A.C. Curcio has a contract funded by the Programme FIRE-POCTEP, a project promoted by the cooperation programme Interreg V-A Spain-Portugal (POCTEP) 2014–2020 and 75% funded by the European Regional Development Fund (ERDF). M. Aranda has a postdoctoral contract, funded by the Programme for the Promotion and Encouragement of Research Activity at the University of Cádiz. All authors have approved each acknowledgment.

**Conflicts of Interest:** The authors declare no conflict of interest.

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
