# Peer review of "Evaluating the Performance of High Spatial Resolution UAV-Photogrammetry and UAV-LiDAR for Salt Marshes: The Cádiz Bay Study Case"

_remotesensing, doi:10.3390/rs14153582_

Round 1
Reviewer 1 Report
Remotesensing- 1779494: High spatial resolution photogrammetry and LiDAR in the Cádiz 2 Bay (SW, Spain): optimizing the application of UAV-techniques 3 to salt marshes
Assessment
This paper presents techniques to develop high-resolution digital models and applied them to an estuary in Spain and the results were assessed and analyzed in the Cadiz Bay marsh system.
General Comments and Suggestions
1. I can say that the major problem of this manuscript is writing and presentation. The structure of sentences, usage of weird words, and some of the graphs need major revisions.
2. Some parts of the results section include discussions that need to be moved to the discussion section. Also, this manuscript needs a better discussion of why having an accurate and high-resolution digital elevation model is important and what an error means in terms of other works that depend on these models (please see and use other studies such as https://doi.org/10.1109/JSTARS.2020.2973490). Also, discuss the transferability of your study around the world and compare it with other studies that have similar approaches.
Specific Comments and Corrections
Page 1-Line 36: see these studies, too: https://doi.org/10.1371/journal.pone.0210134, https://doi.org/10.1371/journal.pone.0058715, https://doi.org/10.1371/journal.pone.0210134
Page 1-Line 40: What’s the relationship between anthropogenic changes and climate change. I think the sentence is wrong and the authors don’t mean it. The problem in writing starts from here till the end of the manuscript.
Page 1-Line 41: along?
Page 1-Line 42: threaten is a verb (before is?). This manuscript needs a lot of work in terms of writing. No one understands it.
Page 1-Line 44: This happens only under low SLR scenarios (see https://doi.org/10.1002/2016EF000385).
Page 2-Line 47: human actions like what? Please be specific. Natural processes like what? Use more citations for your sentences and claims. There is a lot of research about these. This sentence is very vague and general and needs more work and details.
Page 2-Line 51: modeling of what? “may”? “full understanding”? please pay attention to the words you choose. Are you talking about coupled marsh models? Please be specific.
Page 2-Line 59: See this: https://doi.org/10.1109/JSTARS.2020.2973490
Page 2-Line 73-79: paragraph-long sentence. Break it into several sentences.
Page 2- Line 91: This sentence does not make sense. Do you mean use or apply by using the word “work”?
Page 82-Line 93: see https://doi.org/10.1016/j.ecolmodel.2016.01.013
Page 3- Line 108: see https://doi.org/10.3390/rs70403507
Page 3- Line 120: “it” is still necessary…
Page 3- Line 120-123: long sentence. Break it.
Page 3- Line 132-135: long sentence. Break it. Check for all instances like this.
Page 3- Line 147: These features should be shown in Figure 1.
Page 4- Line 158: What do you mean by developed salt marsh? Again I think this is a word-by-word translation and not knowing the exact meaning of the words in English.
Page 4- Line 159: Why Cadiz Bay? Please justify. There are a lot of sites that are easily accessible ? This is not good reasoning. Please find a scientific reason for choosing your site.
Figure 1: Change your Spain map to a simple (not imagery) map and show where the Alboran and Balearic Seas and the Atlantic Ocean are the borders with other countries, Gibraltar, and several important cities. Your zoomed maps should have the lat and long coordinates. The most zoomed one should be a very high-resolution map with captions showing what the red and yellow lines are. Also, pic some of the complexities mentioned in the text and show where they are, etc.
Page 4- Line 165: UAV employed ????
Page 4- Line 165: Please provide some pictures of the UAVs and explain how RTK was installed on them. I am confused if there was an RTK installed on the UAV so that the UAV lands in the marsh and does the measurement or you are talking about just a spatial resolution? Please explain and clarify. Also, make a table and include all the information about the devices in the table.
Page 6- Line 247: explain the software in a sentence and include a citation for that.
Page 6- Line 268: This is so vague. What is the software? Explain and be specific with citations and procedures.
Page 6- Line 270: What do you mean by “favor”? explain and clarify.
Figure 3: What is the connection between the blue box and other boxes?
Page 8- Line 296: Writing problem: To obtain an accurate DEM, it is necessary to classify ….
Page 8- Line 296-299: Long, incorrect, and vague sentence. I am not going to write the manuscript but ask for a major revision for writing the whole manuscript. Good work needs a good presentation otherwise no one can understand it and it will not get the scientific community’s attention.
Page 8- Line 316: allowing excluding?
Page 8- Line 319: departure????
Page 8- Line 324: was à is. The methodology should be present tense. You can use past tense for the results section. Check the whole methodology for this.
Page 10- Line 365: Explain the software. Be specific and name it with citations, etc.
Page 10- Line 376: citation?
Page 10- Line 384: DEM is created… usually?????
Figure 5: Lat long should be added to the map. Make the caption bigger and choose better colors for points and make them shiny so that they can be distinguishable.
Page 11- Line 430: how did you remove them? Was that a controlled fire or something else?
Page 11- Line 440: citation for the software plus a short explanation about it is needed.
Page 13- Line 489: that-> than
Page 15- Line 537: A lot of these errors, which doesn’t allow me to review it. Find all of them and fix them.
Page 17- Line 605-608: This is a discussion, not results. A lot of these instances. Find all of them and move them to the discussion section.
Figure7: Use a blue to red color theme for the map, black and white is not a good choice.
Page 19-Line727: miss?
Reviewer 2 Report
The presented work focuses on application of aereial LiDAR and photogrammetry for salt marshes in the Cadiz Bay. This is an interesting work, but requires some corrections:
- Line 60: fix “theses”
- Line 70: please use “photogrammetric cameras” instead of “photogrammetry camaras”
- Line 171: P1 camera has three kinds of lens, 24mm-35mm-50mm. Please fix the general description. State which you have used.
- Whould be better to add a conclusion chapter by splitting the discussion
- 98 bibliographic references are extremely to much. Please review all citations and thin out this list.
thank you
Reviewer 3 Report
General comments:
It is interesting that the manuscript describes the performance evaluation of the combination of UAV-LiDAR and UAV-photogrammetry for complex coastal wetland classification and modelling. It would be instructive for users in coastal environmental remote sensing. However, it is still difficult to understand the core content.
Specific comments:
1. The title, abstract and introduction section should be rewritten and reframed to provide the concise information, the state-of-the-art research progress, and key scientific or technique issues. I cannot understand the key points, key solutions, or technique improvements from the title and abstract.
2. Unified format. Are there spaces between numbers and units?
3. Figure 1 lacks latitude and longitude. Suggest to provide the field survey photos of the salt marshes with this figure. At least give the main vegetation types or topography information.
4. References were inserted in incorrect formats, such as 2.3.2.
5. The font size in Figure 3 should be increased.
6. Suggest the conclusion as a separate section.
7. Suggest to provide the relative references (e.g., in the journal of Remote Sensing) published in the last five years.
8. The formula of error evaluation should be provided.
9. Please discuss whether the results are consistent with those of others.
10. How to evaluate the combination result? Is it better than single data used? It is unreasonable to use more coarse DEM data to compare the results of this study. Please provide more experiments, analysis, and discussion.
11. What is the spatial resolution of the classification results? Is there any correlation between the overall accuracy and spatial resolution?
12. Does the classification accuracy is dependent with specific data sets? Please discuss.
13. Can this method in the manuscript be generalized to other study areas, or other data sets? Which parameters are sensitive?
14. How to extend this method to a larger scale? If so, the Lidar DEM will be difficult to be acquired. How to tackle this?
Round 2
Reviewer 3 Report
This manuscript can be accepted for publication in present form.